# Recent Advances in the Treatment of Pulmonary Arterial Hypertension

**DOI:** 10.3390/ph15101277

**Published:** 2022-10-17

**Authors:** Naoyuki Otani, Takashi Tomoe, Atsuhiko Kawabe, Takushi Sugiyama, Yasuto Horie, Hiroyuki Sugimura, Takanori Yasu, Takaaki Nakamoto

**Affiliations:** Department of Cardiology, Dokkyo Medical University Nikkyo Medical Center, 632 Takatoku, Nikko 321-2593, Japan

**Keywords:** pulmonary arterial hypertension, vasodilators, therapy, drug targets

## Abstract

Pulmonary arterial hypertension (PAH) is a disease in which stenosis or obstruction of the pulmonary arteries (PAs) causes an increase in PA pressure, leading to right-sided heart failure and death. Basic research has revealed a decrease in the levels of endogenous vasodilators, such as prostacyclin, and an increase in the levels of endogenous vasoconstrictors, such as endothelin, in patients with PAH, leading to the development of therapeutic agents. Currently, therapeutic agents for PAH target three pathways that are selective for PAs: the prostacyclin, endothelin, and nitric oxide pathways. These treatments improve the prognosis of PAH patients. In this review, we introduce new drug therapies and provide an overview of the current therapeutic agents.

## 1. Introduction

Pulmonary arterial hypertension (PAH) is defined as stenosis or obstruction of the pulmonary arteries (PAs) and elevated PA pressure. This results in a gradual increase in pulmonary vascular resistance (PVR), leading to right ventricular failure, impaired quality of life, and death [1]. Thus, PAH treatment is intended to improve hemodynamics by dilating PAs. PAH is primarily treated by vasodilation of the pulmonary vessels. Thirty years ago, it was shown that administering high dosages of calcium channel antagonists to patients with PAH improved survival over a 5-year period [2,3]. However, this treatment was effective only in the few patients who responded to treatment and was not effective in many patients with PAH.

Basic research has revealed a decrease in the levels of endogenous vasodilators, such as prostacyclin, and an elevation in the levels of endogenous vasoconstrictors, such as endothelin, in patients with PAH, leading to the development of therapeutic agents. Currently, therapeutic agents for PAH target three pathways that are selective for PAs: the prostacyclin, endothelin, and nitric oxide (NO) pathways. These treatments have improved the PAH prognosis, which used to be 3 years [4]. In this review, we describe new drug therapies and introduce an outline of the therapeutic agents used at present.

## 2. Definition of Pulmonary Hypertension

Since the 1st World Symposium on Pulmonary Hypertension (WSPH) in 1973, pulmonary hypertension (PH) has been defined as a mean pulmonary arterial pressure (mPAP) ≥ 25 mmHg at rest, measured by right heart catheterization [5]. The most recent data from healthy volunteers indicate that mPAP is 14.0 ± 3.3 mmHg [6]. Therefore, the 6th WSPH Task Force reexamined the criteria for PH, proposing a new pressure value to define an abnormal increase in mPAP > 20 mmHg and the necessity for PVR ≥ 3 Wood units (WU) to define the existence of precapillary PH [7]. One of the reasons this Task Force reduced the mPAP from 25 to 20 mmHg was the advancement in PAH treatment. Further, early intervention for PAH is expected to improve the prognosis.

## 3. Current PAH Medication Therapies

The development of PAH includes three key pathways: the prostacyclin, endothelin, and NO pathways. These pathways are important therapeutic targets and responsible for determining which of the four drug classes will be used: prostacyclin, endothelin receptor antagonists (ERAs), phosphodiesterase type 5 (PDE-5) inhibitors, or soluble guanylate cyclase (sGC) stimulators (Table 1). This dysfunction leads to cellular proliferation, pulmonary vasculature remodeling, and vasoconstriction, leading to increased PVR and, eventually, right heart failure and death. The severity of PAH is classified in accordance with the guidelines offered by the New York Heart Association (NYHA) or World Health Organization (WHO), which link PAH symptoms to physical activity limitations, ranging from functional class (FC) I, where symptoms are mild and do not restrict physical activity, to FC IV, where symptoms are severe and result in an incapability to perform any physical activity [8]. PAH treatment was selected according to NYHA/WHO FCs. In recent years, evidence for initial oral combination therapy has emerged, and prostaglandins, PDE-5 inhibitors, and ERAs have been increasingly used in combination with early treatment [8]. In severe cases, intravenous epoprostenol is considered the first-line treatment of choice, while in moderate cases, multidrug combination therapy using oral and inhaled drugs is used [8]. Sitbon et al., reported the efficacy of a three-drug combination involving epoprostenol, bosentan, and sildenafil in patients with severe PAH of NYHA/WHO FC III or IV [9].

## 4. Prostacyclin Therapy (for Targeting the Prostacyclin Pathway)

Prostaglandin I_2_ (PGI_2_), or prostacyclin, was discovered in 1976 as a vasodilator produced by the vascular endothelium [10]. The binding of prostacyclin to the PGI_2_ receptor activates G proteins and increases intracellular cAMPs, which activates protein kinase A. This leads to platelet aggregation inhibition, smooth muscle relaxation, and PA vasodilation (Figure 1). In PAH patients, the reduced release of prostacyclin [11] and a marked reduction in prostacyclin synthase expression [12] have been reported. However, PAH patients present with low prostacyclin levels, which leads to the constriction of the PAs. Prostacyclin can oppose vasoconstrictive mediators, such as endothelin, which are active in PAH, allowing PA relaxation. Thus, a treatment to replenish depleted prostacyclin is considered rational. Epoprostenol was first used for PAH patients in the early 1980s. The use of prostacyclin and its analogs is a definitive approach to PAH treatment, and these analogs are widely available for clinical therapy to treat PAH. Epoprostenol has been approved as a therapeutic agent for PAH in North America, Japan, and some European countries since the mid-1990s. However, prostacyclin has a short half-life (only minutes), making its use in clinics distressing [13].

Currently, prostacyclin products include epoprostenol and treprostinil for injection, iloprost for inhalation, and beraprost and selexipag for oral administration.

### 4.1. Epoprostenol

Epoprostenol, a synthetic prostacyclin, is a PGI_2_ preparation that has been used since the 1980s for PAH. Epoprostenol has an extremely short half-life and must be administered through a continuous infusion pump and an indwelling central venous catheter, which can trigger life-threatening adverse events, such as pump failure, catheter embolism, and infection. Epoprostenol decreases PVR in a dose-dependent manner and is currently the first-line treatment for patients with PAH in NYHA/WHO FC IV. If treatment with epoprostenol is required, it will be administered starting at a lower dose of 2–4 ng/kg/min and gradually raised to 10–16 ng/kg/min according to adverse drug reactions [14,15]. In a cohort study of 178 patients with PAH, exercise tolerance, hemodynamics, and long-term survival were compared. Overall survival rates at 1, 2, 3, and 5 years were 85%, 70%, 63%, and 55%, respectively. In a multivariate analysis including both baseline variables and measurements, it was demonstrated that 3 months after receiving epoprostenol, a history of right-sided heart failure, the persistence of NYHA functional class III or IV at 3 months, and no more than a 30% decrease in total lung resistance were associated with decreased survival versus baseline [14]. In addition, epoprostenol was demonstrated to increase survival rates in idiopathic PAH patients in randomized controlled trials [16,17]. The survival of patients treated with epoprostenol depends on the severity at baseline as well as the 3-month response to therapy. Epoprostenol can obviate the need for lung transplantation in patients with severe PAH. However, lung transplantation should be considered in patients who do not improve from NYHA/WHO FC III or IV, in those who cannot achieve remarkable hemodynamic success after 3 months of epoprostenol therapy, or both [14]. The optimal epoprostenol dose remains undefined and varies among patients. Hemodynamic improvement is observed even at low doses, and high doses may not always be necessary. Since more treatment options are now available, evidence for treatment at the optimal dose, including combination therapy, is awaited.

### 4.2. Treprostinil

Epoprostenol is chemically unstable at room temperature and should be stored in the refrigerator; however, treprostinil, a prostacyclin analog, is stable at room temperature and neutral pH and has a longer half-life (3–4 h), allowing for continuous subcutaneous administration rather than continuous intravenous administration [18]. This is a unique characteristic of this compound. Treprostinil administration improved the 6 min walk distance (6 MWD), quality of life, pulmonary hemodynamics, and clinical symptoms [19,20]. However, local side effects, such as pain and inflammation at the injection site, are seen in most patients administered treprostinil, which often leads to dose escalation restrictions or treatment discontinuation [14]. Thus, a reliable dosage preparation of treprostinil has been developed for oral administration. This tablet contains the diolamine salt of treprostinil and is a controlled-release formulation designed to provide sustained concentrations of treprostinil in the systemic circulation, allowing for twice-daily administration [21].

### 4.3. Iloprost

Iloprost is a prostacyclin derivative that is inhaled by the patient using a portable nebulizer. It is unaffected by metabolic enzymes, such as cytochrome P450, and has minimal drug interactions. Adherence is an issue because of the short duration of action and the need for six to nine inhalations per day. Olschewski et al., stratified a total of 203 patients according to NYHA FC III or IV and the type of pulmonary hypertension. These patients were then randomized to receive iloprost (*n* = 101) or a placebo (*n* = 102). The primary endpoint was an improvement in the NYHA class and the 6MWD after week 12 by at least one class and at least 10%, respectively, in the absence of clinical deterioration (according to predefined criteria) or death. This combined clinical endpoint was reached in 16.8% of patients receiving iloprost and 4.9% of patients receiving the placebo (*p* = 0.007). The authors found an increase of 36.4 m in the 6MWD in the iloprost group (*p* = 0.004) [22]. Hoeper et al., reported that for 24 PAH patients who received iloprost at a daily dose of 100 or 150 μg for 1 year, the mean distance covered in the 6MWD increased from 278 m at baseline to 363 m after 1 year [23]. Iloprost was generally well tolerated. These data support the conclusion that iloprost is an effective treatment for PAH; however, the long-term efficacy of iloprost inhalation remains unestablished.

### 4.4. Beraprost

Beraprost is a stable, orally active prostacyclin analog with vasodilatory, antiplatelet, and cytoprotective effects, developed in Japan. Unlike epoprostenol, beraprost permits oral ingestion because of its long-lasting activity and produces strong vasodilation and platelet aggregation inhibition. This preparation is characterized by its low cost as a therapeutic agent for PAH. Beraprost may have beneficial effects on survival compared with conventional therapy alone [24]. Galiè et al., reported that for PAH patients in NYHA/WHO FC II or III treated with beraprost for 12 weeks, the 6MWD improved from 362 to 377 m in the beraprost group. The distance walked decreased from 383 to 374 m in the placebo group. The difference in the mean distance walked between the beraprost group and the placebo group (adjusted for baseline values) at week 12 was 25.1 m (95% confidence interval [CI]: 1.8 to 48.3, *p* = 0.036); however, pulmonary hemodynamics and NYHA/WHO FC did not reveal statistically significant improvements [25]. Recently, a long-acting beraprost preparation was developed to provide the sustained release of beraprost sodium for a longer period than the conventional preparation. Kunieda et al., demonstrated that in Japanese PAH patients treated with long-acting beraprost for a 12-week period, the 6MWD showed a significant increase of 33.4 m (95% CI, 13.4 to 53.5) from the baseline measurement. Total pulmonary resistance, mPAP, and PVR decreased by −2.8 mmHg (95% CI, −4.6 to −1.0), −0.92 WU (95% CI, −1.78 to −0.05), and −0.89 WU (95% CI, −1.84 to 0.06), respectively, from the baseline measurements [26]. Currently, beraprost is approved only in Japan and South Korea.

### 4.5. Selexipag

Selexipag is a long-acting non-prostanoid prostacyclin receptor agonist prodrug that can be administered orally. MRE-269, the active form of selexipag, is more selective for the prostacyclin receptor than the prostacyclin analogs beraprost and iloprost, which also have a high affinity for the prostaglandin E3 receptor [27]. The GRIPHON study was an event-driven, phase 3, randomized, double-blind, placebo-controlled trial that enrolled 1156 patients with PAH, who were administered placebo or individualized doses of selexipag [28]. The primary outcome was a composite of death from any cause or a complication related to PAH until the end of the treatment period. Overall, 397 patients had a primary outcome event: 242 patients (41.6%) in the placebo group and 155 patients (27.0%) in the selexipag group. The hazard ratio for a primary outcome event in the selexipag group was 0.60 (95% CI, 0.46 to 0.78; *p* < 0.001). Disease progression and hospitalization accounted for 82% of the events. By the end of the trial, 105 patients in the placebo group and 100 patients in the selexipag group had died of any cause, but there was no significant difference in mortality rates between the two groups (hazard ratio in the selexipag group, 0.97; 95% CI, 0.74 to 1.28; *p* = 0.42). The most common adverse drug reactions with selexipag were headaches, diarrhea, nausea, and jaw pain (Table 1). These were similar to known adverse drug reactions seen with prostacyclin.

## 5. Endothelin Receptor Antagonists (for Targeting the Endothelin Pathway)

Endothelin (ET)-1, an endothelium-derived 21-residue vasoconstrictor peptide, was isolated from the culture supernatant of porcine aortic endothelial cells (ECs) and indicated as one of the most powerful vasoconstrictors in 1988 [29]. After ET-1 was identified, two structurally related peptides were subsequently discovered, ET-2 and ET-3 [30,31]. ET-1, the major isoform of ETs, primarily affects the cardiovascular system and is produced by ECs and various other types of cells, including epithelial cells, macrophages, fibroblasts, cardiomyocytes, and neurons [32]. ET-1 levels were elevated in all PAH cases, regardless of the cause. On the other hand, two isoforms of ET receptors exist, endothelin receptor A (ETRA) and endothelin receptor B (ETRB), which are G-protein-coupled seven-transmembrane receptors [33]. ETRA activation induces continuous vasoconstriction and vascular smooth muscle cell proliferation, while ETRB receptors mediate ET clearance in the lungs and produce NO and prostacyclin [34].

ERAs can improve functional capacity, NYHA/WHO FC, time to clinical worsening, and mortality in PAH patients. Currently, the ERAs approved for PAH include bosentan, ambrisentan, and macitentan. However, there is still insufficient evidence as to which of these ERAs is most effective in PAH.

### 5.1. Bosentan

Bosentan is a non-peptidic ERA, specifically a dual oral ETRA and ETRB antagonist. Bosentan (the treatment of PAH is intended) has been assessed in five randomized controlled trials (Pilot, Bosentan Randomized Trial of Endothelin Antagonist Therapy-1 (BREATHE-1), BREATHE-2, BREATHE-5, and EARLY) that have demonstrated improvements in exercise capacity, NYHA/WHO FC, pulmonary hemodynamics, and time to clinical worsening [35,36,37,38,39]. Thirty-two patients with pulmonary hypertension and NYHA/WHO FC III were randomly assigned to either the bosentan or the placebo group for a minimum of 12 weeks in a pilot study. The primary endpoint was an improvement in exercise capacity. The 6MWD improved by 70 m at 12 weeks compared with baseline values in the bosentan group, whereas it worsened by 6 m in the placebo group (95% CI, 12 to 139, *p* = 0.021). This improvement was maintained for at least 20 weeks. Patients who received bosentan had significantly improved exercise capacity and similarly improved mPAP, cardiac output, and PVR [35]. In BREATHE-1, 213 PAH patients in NYHA/WHO FC III or IV were randomly assigned to receive placebo or bosentan, which also increased the 6MWD; after 16 weeks, the 6MWD increased by 36 m in the bosentan group, while it decreased by 8 m in the placebo group (95% CI, 21 to 67, *p* <0.001). Both doses of bosentan showed a significant therapeutic effect, but the placebo-corrected improvement was more pronounced in the 250 mg b.i.d. group than in the 125 mg b.i.d. group (54 m and 35 m, respectively). No dose–response relationship could be confirmed for efficacy, however [36]. Bosentan was approved for the treatment of PAH in North America in 2001, Europe in 2002, and Japan in 2005. The bosentan regimen was started at 62.5 mg twice daily and increased to 125 mg twice daily after 1 month. In the EARLY study, a double-blind, placebo-controlled, multicenter study, PAH patients with WHO FC II less than 80% of the normal predicted level or with a 6MWD of less than 500 m, associated with a Borg dyspnea index of 2 or more, were enrolled, and 185 patients were assigned to receive either bosentan (*n* = 93) or placebo (*n* = 92). Primary endpoints were PVR and the 6MWD at 6 months. Geometric mean PVR was 83.2% (95% CI, 73.8 to 93.7) of the baseline level in the bosentan group and 107.5% (97.6 to 118.4) of the baseline level in the placebo group (treatment effect −22.6%, 95% CI, −33.5 to −10.0; *p* < 0.0001) at 6 months. The mean 6MWD increased from baseline in the bosentan group (11.2 m, 95% CI, −4.6 to 27.0) and decreased in the placebo group (−7.9 m, −24.3 to 8.5), with a mean treatment effect of 19.1 m (95% CI, 3.6 to 41.8; *p* = 0.0758) [39]. Adverse drug reactions of bosentan included an increase in liver enzymes and teratogenicity [39]. Liver injury was observed to be dose-dependent. However, the condition improved after dose reduction or discontinuation, suggesting that the injury was reversible. This injury is also seen in patients administered other ERAs, such as ambrisentan and sitaxsentan (Table 1).

### 5.2. Ambrisentan

Ambrisentan is a strong oral ETRA-selective antagonist characterized by high bioavailability and a long half-life. The drug can be administered once daily due to its half-life of 9–15 h. In a double-blind, dose-ranging study, 64 PAH patients were randomized to receive 1, 2.5, 5, or 10 mg of ambrisentan once daily for 12 weeks, followed by 12 weeks of open-label ambrisentan. The primary endpoint was an improvement in the 6MWD compared with the baseline values. Ambrisentan increased the 6MWD (+36.1 m, *p* < 0.0001) at 12 weeks, and similar significant increases for each dose group were noted [40]. ARIES (Ambrisentan in Pulmonary Arterial Hypertension, Randomized, Double-Blind, Placebo-Controlled, Multicenter, Efficacy Study)-1 and -2 were double-blind, placebo-controlled trials that randomized 202 and 192 PAH patients, respectively, to placebo or ambrisentan for 12 weeks. In ARIES-1, the mean placebo-corrected treatment effects were 31 m (*p* = 0.008) and 51 m (*p* < 0.001) for 5 and 10 mg ambrisentan, respectively. In ARIES-2, the mean placebo-corrected treatment effects were 32 m (*p* = 0.022) and 59 m (*p* < 0.001) for 2.5 and 5 mg ambrisentan, respectively. In this trial, serum aminotransferase concentrations greater than three times the upper normal limits were not observed in ambrisentan-treated patients [41]. Ambrisentan has been approved for PAH patients with NYHA/WHO FC II and III. The incidence of abnormal liver function tests with ambrisentan use ranges from only 0.8% to 3% [42], but an increased incidence of peripheral edema has been reported [42]. Thus, the safety of ambrisentan is considered superior to that of other ERAs.

### 5.3. Macitentan

Macitentan, a dual ETRA/ETRB antagonist, was designed by modifying the structure of bosentan to enhance its efficacy and safety. In the event-driven SERAPHIN (Study with an ERA in PAH to Improve Clinical Outcome) trial, 742 PAH patients with NYHA/WHO FC II, III, or IV were administered 3 or 10 mg of macitentan compared with placebo for an average duration of 100 weeks. The primary endpoint was defined as the time from the initiation of treatment to the first occurrence of a composite endpoint of death, atrial septostomy, lung transplantation, the initiation of treatment with intravenous or subcutaneous prostacyclin, or the worsening of PAH. A total of 250 patients were randomized to the placebo group, 250 were randomized to the 3 mg macitentan dose group, and 242 were randomized to the 10 mg macitentan dose group. A primary endpoint occurred in 46.4%, 38.0%, and 31.4% of the patients in these groups, respectively. The hazard ratio for the 3 mg dose of macitentan was 0.70 (97.5% CI, 0.52 to 0.96; *p* = 0.01), compared to that of the placebo. For the 10 mg dose of macitentan compared to that of the placebo, the hazard ratio was 0.55 (97.5% CI, 0.39 to 0.76; *p* < 0.001) [43]. Macitentan was approved by the FDA in 2013 and the European administration and Japan in 2015.

### 5.4. Sitaxsentan

Sitaxsentan is an ERA characterized by high oral bioavailability, a long duration of action, and high specificity for ETRA. Sitaxsentan could benefit PAH patients by inhibiting the vasoconstrictor effects of ET-A while preserving the vasodilator response and ET clearance functions of ETRB in the lungs. The efficacy and safety of sitaxsentan were also assessed, which showed improvements in exercise capacity and hemodynamics [44,45,46]. While hepatotoxicity is a recognized side effect of treatment with bosentan, sitaxsentan was withdrawn from the market because the liver injuries suffered by the two patients were severe [47].

## 6. Phosphodiesterase Type 5 Inhibitors and Soluble Guanylate Cyclase Stimulators (Targeting the Nitric Oxide Pathway)

NO is mainly produced and released by vascular endothelial cells and acts on vascular smooth muscle cells, activating soluble guanylate cyclase to produce cyclic guanosine monophosphate (cGMP) from guanosine triphosphate (GTP). The pathogenesis of PH involves impaired NO synthesis and impaired signaling through the NO-sGC-cGMP pathway. PDE-5 inhibitors, such as sildenafil, tadalafil, and vardenafil, inhibit cGMP-degrading enzymes, resulting in vasodilation through the NO/cGMP pathway and the inhibition of vascular smooth muscle cell growth in the lung (Figure 1). In contrast, sGC stimulators promote cGMP production and may be effective even in conditions of depleted endogenous NO. PDE-5 inhibitors and sGC stimulators are both PH drugs that target the NO-cGMP pathway, a signaling pathway that is crucial for the regulation of pulmonary vascular tonus [42].

### 6.1. Sildenafil

Sildenafil is a selective oral inhibitor of cGMP-specific PDE-5, an enzyme that is abundant in both the lung and penile tissues. Sildenafil is widely used to treat erectile dysfunction to dilate penile arteries [48]. Sildenafil also inhibits PDE-5, which is abundant in lung tissues, prevents the degradation of cGMP, and dilates pulmonary vascular beds. The approved dose of sildenafil is 20 mg three times daily. In the Sildenafil Use in Pulmonary Arterial Hypertension (SUPER) Study, a randomized, placebo-controlled, double-blind, comparative study lasting 12 weeks, PAH patients with NYHA/WHO FC II or III were treated with sildenafil. The 6MWD increased from baseline values in all sildenafil groups. The placebo-corrected mean treatment effect was 45 m (+13.0%), 46 m (+13.3%), and 50 m (+14.7%) for sildenafil administered at 20, 40, and 80 mg, respectively (*p* < 0.001 for all). All sildenafil doses decreased mPAP (*p* = 0.04, *p* = 0.01, and *p* < 0.001, respectively) and improved WHO FC (*p* = 0.003, *p* < 0.001, and *p* < 0.001, respectively) [49]. Patients who completed the SUPER-1 study were eligible to participate in a subsequent, open-label uncontrolled extension study (SUPER-2). This study continued until the last patient completed 3 years of sildenafil therapy. Sildenafil single-agent therapy was well tolerated over the long term in this study. Most patients in the SUPER-1 trial improved or maintained their NYHA/WHO FC or 6MWD. In contrast, the placebo group (for 12 weeks, they received placebo in SUPER-1 and then received sildenafil in SUPER-2) had lower longer-term survival than the groups receiving sildenafil treatment from the start of SUPER-1 [50]. These results highlight the importance of early PAH treatment.

### 6.2. Tadalafil

Tadalafil, an orally administered long-acting agent and selective inhibitor of PDE-5, increases cGMP levels in the NO pathway. It has been approved for the treatment of PAH and erectile dysfunction. In the Pulmonary Arterial Hypertension and Response to Tadalafil (PHIRST) Study, a double-blind, placebo-controlled study, 405 patients with PAH were randomly assigned to receive placebo or tadalafil 2.5, 10, 20, or 40 mg orally once daily. Overall, the mean placebo-corrected treatment effect was 33 m (95% CI, 15 to 50 m). Tadalafil increased the 6MWD at 16 weeks, as evaluated by the primary endpoint, in a dose-dependent manner, but only the 40 mg dose of tadalafil met the level of statistical significance. Additionally, 40 mg of tadalafil improved the time to clinical worsening, the incidence of clinical worsening, and health-related quality of life in this study, but the changes in NYHA/WHO FC were not statistically significant. Thus, these results indicate that 40 mg of tadalafil was the optimal dose for PAH patients [51]. Patients completing the PHIRST-1 study for 16 weeks were subsequently enrolled in the double-blind, 52-week, uncontrolled extension study (PHIRST-2). The 357 eligible patients received tadalafil 20 or 40 mg once daily, and 293 patients completed this study. In patients administered either 20 or 40 mg of tadalafil, the increase in the 6MWD was sustained for up to 52 additional weeks of treatment in PHIRST-2, as in the PHIRST study [52]. The adverse drug reactions of tadalafil were reported as headaches, myalgia, and flushing (Table 1).

Coyle et al., compared the cost-effectiveness of single-agent therapy with oral PAH-specific therapies, including bosentan, ambrisentan, riociguat, tadalafil, sildenafil, and supportive care, as initial therapy for PAH patients with NYHA/WHO FC II or III [53]. The results indicated that initiation therapy for PAH with sildenafil is likely the most cost-effective strategy.

### 6.3. Vardenafil

Vardenafil is a PDE-5 inhibitor approved for the treatment of erectile dysfunction in 2005 and can be administered twice daily. Vardenafil is more potent and selective than sildenafil in inhibiting PDE-5. Vardenafil is similarly effective to sildenafil in the treatment of erectile dysfunction. The Efficacy and Safety of Vardenafil in the Treatment of Pulmonary Arterial Hypertension (EVALUATION) was a randomized, double-blind, placebo-controlled, multicenter study that examined the effects of 12 weeks of oral 5 mg of vardenafil twice daily on exercise capacity, pulmonary hemodynamics, and clinical symptoms. Administering vardenafil 5 mg twice daily to 66 treatment-naive PAH patients demonstrated improvements in exercise capacity, hemodynamics, and time to clinical worsening. The median 6MWD increased by 59 m in the vardenafil group, whereas it decreased by 10 m in the placebo group. Therefore, the mean placebo-corrected treatment effect of vardenafil was 69 m (95% CI, 41 to 98 m; *p* < 0.001) [54]. However, vardenafil is not yet approved for PAH at present.

### 6.4. Riociguat

Riociguat is a therapeutic class of sGC stimulators. Riociguat has a dual mechanism of action, acting synergistically with endogenous NO and directly stimulating sGC independently of NO availability. Pulmonary Arterial Hypertension Soluble Guanylate Cyclase-Stimulator Trial 1 (PATENT-1) was a randomized, double-blind, phase 3 study of 443 patients with symptomatic PAH receiving either placebo or riociguat. In this study, riociguat significantly increased exercise capacity after 12 weeks, the primary endpoint in PAH patients. The 6MWD increased from baseline values by a mean of 30 m in the 2.5 mg riociguat group and decreased by a mean of 6 m in the placebo group (least-squares mean difference, 36 m; 95% CI, 20 to 52; *p* < 0.001). Riociguat also improved the secondary efficacy endpoints, including pulmonary hemodynamics, NYHA/WHO FC, and time to clinical worsening. The most common serious adverse event observed in this study was syncope, which was 4% in the placebo group, 1% in the group receiving 2.5 mg riociguat, and 0% in the group receiving 1.5 mg of riociguat [55].

## 7. Initial Dual Oral Combination Therapy

Various specific PAH agents that act on three pathways, the prostacyclin, endothelin, and NO pathways, have been developed and were reported to improve various endpoints in several clinical trials. However, despite treatments with such agents, the progression of PAH is frequently observed. Therefore, multidrug regimens that simultaneously target different targets seem to be an attractive concept [56,57]. Combination therapy with different mechanisms of action to enhance further clinical benefits can be a new option in the treatment of PAH.

The standard treatment of PAH has classically been monotherapy, and several agents for other mechanisms are added sequentially when the response to treatment is not successful. The current recommendation for initial oral combination therapy in the treatment algorithm is based on the following large clinical trial: The Ambrisentan and Tadalafil in Patients with Pulmonary Arterial Hypertension (AMBITION) trial, which examined the efficacy of initial combination therapy with ambrisentan and tadalafil. The primary endpoint in this trial was the first event of clinical failure, which was defined as the first occurrence of a composite of death, hospitalization for worsening pulmonary arterial hypertension, disease progression, or an unsatisfactory long-term clinical response. In patients with PAH who had not received previous treatment, initial combination therapy with ambrisentan and tadalafil resulted in a 50% lower risk of clinical failure events than the risk with ambrisentan or tadalafil monotherapy [58]. This trial supports the rationale for targeting multiple pathways in PAH and demonstrates that combination therapy could be beneficial in the initial phase of treatment. Non-responders to acute vasoreactivity testing who were at low or intermediate risk were suggested to be treated with initial oral combination therapy with an ERA and a PDE-5 inhibitor in the 2015 European Society of Cardiology and the European Respiratory Society PH guidelines.

Furthermore, combination therapy with tadalafil and ambrisentan in de novo NYHA/WHO-FC II and III PAH patients has been assigned a grade 1 recommendation [59].

Sitbon et al., also studied the effects of initial combination therapy with ERAs and PDE-5 inhibitors. This retrospective study of 97 treatment-naïve patients with PAH in NYHA/WHO FC III or IV investigated initial dual oral combination therapy with an ERA (bosentan or ambrisentan) and a PDE-5 inhibitor (sildenafil or tadalafil). Simultaneous combined ERA and PDE-5 inhibitor therapy may offer benefits for treatment-naive patients with PAH [60].

Sulica et al., examined the effectiveness of initial macitentan and riociguat combination therapy in an analysis of 15 consecutive patients and found that the combination of macitentan and riociguat was effective [61]. However, the limitations in this study were the retrospective design, small sample size, and patients with heterogeneous etiologies of PAH.

Post hoc analyses of the GRIPHON trial investigated the efficacy, safety, and tolerability of selexipag compared with placebo in a subgroup of patients receiving an ERA and PDE-5 inhibitor at baseline. The addition of selexipag to double combination therapy provided an incremental benefit similar to that observed among the overall patients in the GRIPHON trial. Triple combination therapy targeting three etiologic pathways is also promising; however, prospective randomized trials that clearly indicate the long-term clinical benefits of the combination are lacking [62].

## 8. Development of New Therapeutic Agents

Continuous intravenous infusion of epoprostenol, a pulmonary vasodilator, has been used in the treatment of PAH since the 1980s. Over the past 30 years, the treatment of PAH has changed substantially, with four classes of agents currently being utilized: prostacyclins, ERAs, PDE-5 inhibitors, and sGC stimulants. Consequently, the prognosis of PAH has significantly improved.

The goals of PAH treatment are to improve subjective symptoms and life expectancy. However, the infrequency of PAH makes it difficult to conduct large-scale clinical trials. As we have discussed, the 6MWD has often been used as the primary endpoint in clinical trials. However, time to clinical worsening has recently been used more frequently. In order to clarify the effect of treatment in a small number of cases, an evaluation index that can accurately identify the pathophysiology of the disease is needed. The ideal evaluation index would be a surrogate marker whose value fluctuates to reflect the effect of treatment and correlates with life expectancy. Surrogate markers have been used in many studies; however, to date, no new or clinically useful evaluation index has emerged. The development of useful evaluation indices is also considered necessary to quantify the effect of treatment.

Basic studies revealed that PAH is associated with the progression of remodeling lesions, such as tunica media thickening, afferent neointimal proliferation, and plexiform lesions [63]. This indicates that hemodynamic compromise and obstructive lesions are involved in the formation of PAH lesions. Some reports have implicated the proliferation of abnormal neoplastic cells in this obstructive lesion, suggesting that this tumor-like growth may be a potential target for future research. Conversely, hemodynamic stress due to increased pulmonary vascular resistance may induce inflammation and cause obstructive lesions [64]. It is assumed that these factors interact to create a vicious cycle in these patients.

Our ability to treat PAH has improved dramatically. However, cases where drug therapy is ineffective remain, leading to lung transplantation as a last resort. Therefore, PAH remains an area of high unmet medical need for which therapeutic agents with new mechanisms of action are expected. While the current focus of therapy is to dilate blood vessels in the lungs, new approaches may include the suppression of inflammation and cell proliferation. We have discussed those new drugs, categorizing them into those targeting known drug targets, so-called drug repositioning, and those targeting new drug targets (Table 2). 

### 8.1. Drugs for Already-Known Targets

#### 8.1.1. Tyrosine Kinase Inhibitors

The three amino acid residues that are phosphorylated in proteins are tyrosine, serine, and threonine, and the enzymes that phosphorylate these substrates are generally called kinases. Tyrosine kinases selectively phosphorylate tyrosine residues. Tyrosine kinases are generally responsible for inducing cell proliferation. Imatinib is an oral anticancer agent developed to target BCR-ABL tyrosine kinase for the treatment of chronic myeloid leukemia. Disordered pulmonary vascular remodeling occurs in the pathophysiology of PAH, as in cancer cells. Platelet-derived growth factor (PDGF) promotes the proliferation of PA smooth muscle cells, leading to the obstruction of small PAs [65]. The inhibitory effects of imatinib on PDGF receptors indicate that imatinib may be effective for PAH. A phase 2, randomized, double-blind, placebo-controlled trial was conducted to evaluate the safety, tolerability, and efficacy of imatinib in 59 PAH patients. Although imatinib appeared to be safe and well tolerated over 6 months, it did not significantly improve in the 6MWD as a primary outcome. However, hemodynamic improvements in this study allow imatinib to be potentially effective as an adjunctive therapy. Imatinib in Pulmonary Arterial Hypertension, a Randomized, Efficacy Study (IMPRES), is a phase 3, multicenter, randomized, double-blind, placebo-controlled trial that lasted 24 weeks, which showed that imatinib improved exercise capacity and pulmonary hemodynamics in patients with advanced PAH despite receiving combination therapy with two or more drugs for PAH. Most adverse events reported in this trial were the same as those previously observed in patients with other indications for imatinib, such as nausea, peripheral edema, diarrhea, vomiting, and periorbital edema. Study drug discontinuations were comparatively higher in this trial than in previous studies with imatinib. In particular, the incidence of subdural hematomas was unexpectedly high (4.2%). However, the cause–effect relationship and mechanism are unknown, and further research is required [66]. As a result, regulatory consideration of PAH indications for imatinib has been halted. On the contrary, tyrosine kinase inhibitors, predominantly dasatinib, have been known to trigger PAH. In addition to the known association between dasatinib and PAH, several tyrosine kinase inhibitors, including ponatinib, bosutinib, and lapatinib, have recently been reported to cause PAH [67]. Although the pathogenesis in these cases is unknown, there is a possibility of drug-induced PAH caused by tyrosine kinase inhibitors. Thus, it is difficult to develop an agent for this PDGF pathway.

#### 8.1.2. Serotonin Receptor Antagonists

Serotonin, also termed 5-hydroxytryptamine (5-HT), is a monoamine neurotransmitter that regulates dopamine and noradrenaline in the brain. 5-HT acts as a growth factor for PA smooth muscle cells, possibly contributing to the pathophysiology of PAH development and progression [68,69]. 5-HT receptors are classified into seven types, and 5-HT_1B_, 5-HT_2A_, 5-HT_2B_, and 5-HT_7_ receptors are expressed in smooth muscle and endothelial cells of the pulmonary vasculature. The plasma levels of 5-HT are increased in PAH patients. Thus, the blockade of 5-HT receptors can be a useful strategy for PAH therapy [70]. Terguride, a potent antagonist of the 5-HT_2B_ and 5-HT_2A_ receptors, is well tolerated and clinically approved for the treatment of ovulation disorders. However, terguride failed to demonstrate clinical efficacy in a phase 2 study for PAH patients [71].

#### 8.1.3. Rho-Kinase Inhibitors

In 1995, Rho kinase was identified as an effector of Rho [72]. The serine–threonine protein kinase Rho kinase family, a downstream effector of the small GTPase Rho, regulates smooth muscle contraction, actin cytoskeleton organization, cell adhesion and motility, cytokinesis, and gene expression. The activation of Rho kinase plays a major role in the pathogenesis of several cardiovascular diseases, including PAH [73], and is thus a potential new therapeutic target for PAH [74]. Fasudil is one of the most commonly used Rho-kinase inhibitors and is clinically applied as an agent for inhibiting vasospasm after arachnoid hemorrhage. Fukumoto et al., performed a double-blind, placebo-controlled clinical trial as a pilot efficacy trial (phase IIa trial) [75]. The results indicated that a 3-month treatment with oral fasudil significantly increased the cardiac index in 30 PAH patients. Moreover, serum levels of Rho-kinase inhibitor correlated with both an increase in the cardiac index and a decrease in mPAP. Notably, fasudil had no serious side effects in this study. However, there are no long-term efficacy and safety trials of fasudil.

#### 8.1.4. Ranolazine

Ranolazine was approved for the treatment of stable angina pectoris in 2006 and has excellent safety and tolerability in patients with ischemic heart disease. Ranolazine can reduce calcium overload in failing cardiomyocytes by suppressing the inward late sodium current and inhibiting fatty acid oxidation [76]. Khan et al., performed a 3-month, prospective, open-label pilot study in 11 patients with symptomatic PAH. Patients receiving ranolazine for 3 months showed improvements in clinical symptoms in NYHA/WHO FC and the RV structure and function on echocardiography; however, it did not alter invasive hemodynamic parameters [77].

#### 8.1.5. Tocilizumab

PAH has multifactorial pathobiology, and it has been documented that autoimmunity is important in the pathophysiology of PAH [78]. The analysis of 60 patients with PAH showed significantly higher levels of interleukin-1-beta (IL-1β), IL-2, IL-4, IL-6, IL-8, IL-10, and IL-12p70 and tumor necrosis factor-α than their levels in healthy controls. Furthermore, high levels of IL-6, IL-8, IL-10, and IL-12p70 predicted survival in patients [79]. IL-6 is a multifunctional proinflammatory cytokine that is known to be elevated in the serum of PAH patients. In particular, the IL-6/IL-21-signaling axis is thought to play a critical role [80]. Tocilizumab, an IL-6 receptor antagonist, is used clinically primarily for chronic rheumatoid arthritis and has been shown to be safe and effective. TRANSFORM-UK was an open-label study of tocilizumab in patients with PAH [81]. The primary endpoints were the incidence and severity of adverse events within 6 months after receiving intravenous tocilizumab. The secondary endpoints were the 6MWD, NT-proBNP, the NYHA/WHO FC assessment of patient-reported symptoms, and quality of life. This study was performed between 2016 and 2018 in the United Kingdom. 

### 8.2. Drugs with New Targets

#### Sotatercept

As the pathogenesis of PAH has been clarified, several new factors associated with the pathogenesis of PAH have been identified. It has long been well known that PAH has a genetic component and that disease clustering occurs in some families. Mutations in the bone morphogenetic protein receptor 2 (BMPR2) gene are the most common cause of hereditary PAH. Mutations in activin A receptor type II-like 1, endoglin, caveolin-1, and members of the Smad family, including mothers against decapentaplegic homolog (SMAD)1, SMAD4, and SMAD9, have been identified [82,83]. Patients with mutations in transient receptor potential canonical 6 (TRPC6), a voltage-independent Ca^2+^ channel, may be at an increased risk of developing idiopathic PH [84]. Potassium Two Pore Domain Channel Subfamily K Member 3 (KCNK3) or Rho kinase may be associated with the pathogenesis of PAH [85,86]. 

Mutations in BMPR2, a member of the transforming growth factor-β (TGF-β) superfamily, are a major factor underlying heritable PAH. Sotatercept acts as a ligand trap for members of the TGF-β superfamily, thus restoring the balance between the growth-promoting activin growth differentiation factor pathway and growth-inhibiting BMP pathway [87]. In the PULSAR trial, 106 patients with PAH treated with sotatercept for 24 weeks had significantly reduced PVR, the primary endpoint, compared with those receiving the placebo. In addition, receiving sotatercept improved exercise capacity, as evaluated by the 6MWD, and decreased N-terminal pro-brain natriuretic peptide (NT-proBNP) levels [87]. However, the long-term efficacy of sotatercept has not been well established. The Study of Sotatercept for the Treatment of Pulmonary Arterial Hypertension (STELLAR) is a phase 3, randomized, double-blind, placebo-controlled study to assess the efficacy and safety of sotatercept when added to conventional therapy for PAH, which is in progress. Patients will be randomized to receive either a subcutaneous injection of 0.7 mg/kg sotatercept once every 3 weeks or placebo. The primary efficacy endpoint in the study is exercise capacity, as assessed by the 6 MWD at 24 weeks after the initiation of treatment. The trial will be completed in April 2023.

## 9. Conclusions

This review shows an outline of current drug therapies for PAH and summarizes the latest topics. PAH pathophysiology is heterogeneous and multifactorial. Understanding the underlying mechanisms of PAH has led to the development of new treatments, such as prostacyclin, ERAs, and PDE-5 inhibitors. However, some patients with PAH are still refractory to treatment, and it is hoped that new treatment strategies will be developed in the future.

## Figures and Tables

**Figure 1 pharmaceuticals-15-01277-f001:**
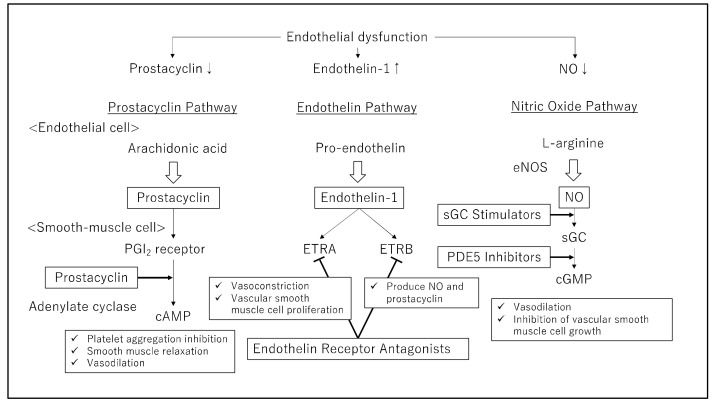
Three pathways of targeted therapeutic agents for pulmonary artery hypertension. Current therapeutic agents are targeted at correcting endothelial dysfunction by inhibiting the endothelin pathway and enhancing the prostacyclin or nitric oxide (NO) pathway. PGI2: prostaglandin I2; ETRA: endothelin A receptor; ETRB: endothelin B receptor; eNOS: endothelial isoform of nitric oxide synthase; sGC: soluble guanylate cyclase, PDE5: phosphodiesterase type 5, cAMP: cyclic adenosine monophosphate; cGMP: cyclic guanosine monophosphate.

**Table 1 pharmaceuticals-15-01277-t001:** First-line drugs for targeting endothelial dysfunction for PAH.

Target-Based Actions	Drug Name	Dosage	Adverse Effects
Prostacyclins	Epoprostenol	Initial: 2–4 ng/kg/min IV infusion pump. Titrate by 1–2 ng/kg/min every 15 min or longer and gradually raise to 10–16 ng/kg/min according to adverse drug reactions.	>10% Flushing jaw pain Headache Myalgia Diarrhea, nausea, and vomiting Flu-like symptoms Eczema Rash Urticaria Hypotension Anxiety Nervousness
	Treprostinil	<Injectable> Initial: 1.25 ng/kg/min continuous SC/IV infusion (0.625 ng/kg/min if not tolerated). Titrate by no more than 1.25 ng/kg/min every 1 week in first 4 weeks, then no more than 2.5 ng/kg/min every 1 week. <Extended-release tablets> Initial: 0.125 mg PO TID or 0.25 mg PO BID. Titrate by 0.125 mg TID or 0.25 or 0.5 mg BID, not more frequently than every 3–4 days.	>10% Infusion site reaction, pain Headache Nausea Diarrhea Vasodilation Jaw pain Rash 1–10% Edema Pruritis Hypotension
	Iloprost	Initial: 2.5 μg inhaled by a nebulizer, then 5 μg subsequent doses 6–9 times/day.	>10% Flushing Cough Hypotension Nausea Headache Jaw pain Trismus 1–10% Palpitations <1% Bronchospasm Supraventricular tachycardia
	Beraprost	Initial: 20 μg PO TID, to highest tolerated dose; not to exceed 180 μg TID.	Bleeding tendencies Symptomatic hypotension Flushing Headache Rash Increased liver aminotransferases
	Selexipag	Initial: 0.2 mg PO BID. Increase dose by 0.2 mg BID, at weekly intervals, to highest tolerated dose; not to exceed 1.6 mg BID.	>10% Headache Diarrhea Jaw pain Nausea and vomiting Pain in extremity Myalgia Flushing Arthralgia Rash 1–10% Anemia Frequency not defined TSH reduced Symptomatic hypotension
Endothelin receptor antagonists	Bosentan	<40 kg: Maintain dose at 62.5 mg PO BID. >40 kg: 62.5 mg PO BID for 4 weeks and then increased to maintenance dosage 125 mg PO BID; not to exceed 250 mg BID.	>10% Anemia Respiratory tract infection Headache Edema Nasopharyngitis Flushing Hypotension 1–10% Hypotension Increased liver aminotransferases
	Ambrisentan	Initiate treatment at 5 mg PO QD, as needed and tolerated, not to exceed 10 mg.	>10% Peripheral edema Headache
	Macitentan	10 mg PO QD	>10% Nasopharyngitis Headache Anemia Bronchitis 1–10% Increased liver aminotransferases
	Sitaxsentan	Withdrawn from the market	
Phosphodiesterase type 5 inhibitors	Sildenafil	5 mg or 20 mg PO TID; administer 4–6 h apart	>10% Headache Flushing Dyspepsia Abnormal vision
	Tadalafil	40 mg PO QD	>10% Headache Myalgia Respiratory tract infection Nasopharyngitis Dyspepsia Flushing Back pain Nausea
	Vardenafil	Vardenafil is not yet approved for PAH at present.	
Soluble guanylate cyclase stimulators	Riociguat	Initial dose: 1 mg PO TID; consider 0.5 mg PO TID if patient may not tolerate hypotensive effect. If systolic blood pressure > 95 mmHg and no symptoms of hypotension, up-titrate dose by 0.5 mg PO TID, with dose increases no sooner than 2 weeks apart to highest tolerated dose. Maintenance dose: not to exceed 2.5 mg PO TID. If symptoms of hypotension occur, decrease dose by 0.5 mg TID.	>10% Headache Dyspepsia and gastritis Dizziness Nausea and vomiting Diarrhea 1–10% Hypotension Hemoptysis

**Table 2 pharmaceuticals-15-01277-t002:** Development of new drugs for PAH.

	Target-Based Actions	Drug Name	Administration Route	Development Status
Drugs for already-known targets	Tyrosine kinase inhibitors	Imatinib	Oral administration	Phase 3 (no development)
	Serotonin receptor antagonists	Terguride	Oral administration	Phase 2 (no development)
	Rho-kinase inhibitors	Fasudil	Oral administration	Phase 2
	Inhibition of I_Na_ and fatty acid oxidation	Ranolazine	Oral administration	Phase 3
	IL6 receptor antagonists	Tocilizumab	Injection (intravenous administration)	Phase 3
Drugs in new targets	Ligand trap for TGF-β	Sotatercept	Injection (subcutaneous injection)	Phase 3

## Data Availability

Not applicable.

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
