# Peer review of "Recent Advances in the Treatment of Pulmonary Arterial Hypertension"

_pharmaceuticals, 2022, doi:10.3390/ph15101277_

Round 1

Reviewer 1 Report

The review is very well written, crisp and to the point. However, I would request the authors to consider the following comments

1. Although this is a narrative review, kindly add a section on methods. As to where the references articles are taken from, what search terms were used etc. 

2. It will be good if a picture showing the various etiological pathways causing PAH are shown and mark where the drug classses act with their mechanism of action

3 Please consider a summary table with the drugs, key PK parameters that translate to clinical care and ADRs. May also consider adding the recommended doses

4. Although evidence from literature is quoted, they are generic. No values / numbers are mentioned. For example, "composite endpoint of death or a complication related to PAH was significantly improved in the selexipag group" Please give exact numbers/ values for the readers to make informed interpretations. 

5. The unmet need for new drugs mentioned may be given in detail under a separate heading before the start of the development of new drugs

6. The new drugs may be classified as drugs in pipeline for already know targets and durgs in pipeline in new targets. 

Author Response

Response to Reviewer 1 Comments

Thank you for inviting us to submit a revised draft of our manuscript, entitled, “Recent Advances in the Treatment of Pulmonary Arterial Hypertension,” to Pharmaceuticals. We also appreciate the time and effort you and each of the reviewers have dedicated to provide insightful feedback on ways to strengthen our paper. We have attached the revised manuscript incorporating your suggestions for further consideration. We also hope that our edits and the point-by-point responses we provided below satisfactorily address all the issues and concerns you and the reviewers have noted.

Point 1: Although this is a narrative review, kindly add a section on methods. As to where the references articles are taken from, what search terms were used etc.  

Response 1: You have raised an important point; however, we believe that it is difficult to add a section on methods in a narrative review.

Point 2: It will be good if a picture showing the various etiological pathways causing PAH are shown and mark where the drug classses act with their mechanism of action

Response 2: Thank you for pointing this out. We have included a new figure (Figure 1) to illustrate the various etiologic pathways that cause PAH and have indicated the mechanism of action of the different classes of drugs.

Point 3: Please consider a summary table with the drugs, key PK parameters that translate to clinical care and ADRs. May also consider adding the recommended doses.

Response 3: Thank you for pointing this out. We have included a new figure (Figure 1) to illustrate the various etiologic pathways that cause PAH and have indicated the mechanism of action of the different classes of drugs.

Point 4: Although evidence from literature is quoted, they are generic. No values / numbers are mentioned. For example, "composite endpoint of death or a complication related to PAH was significantly improved in the selexipag group" Please give exact numbers/ values for the readers to make informed interpretations.

Response 4: We agree with you and have incorporated this suggestion throughout our paper and included the exact values / numbers.

Point 5: The unmet need for new drugs mentioned may be given in detail under a separate heading before the start of the development of new drugs. The unmet need for new drugs mentioned may be given in detail under a separate heading before the start of the development of new drugs.

Response 5: Thank you for your suggestion. As per your suggestion, we have added the unmet need for new drugs before the start of the development of new drugs (p. 13, lines 416-447).

Point 6: The new drugs may be classified as drugs in pipeline for already know targets and durgs in pipeline in new targets.

Response 6: Thank you for this helpful comment. We agree with you and have incorporated this suggestion throughout our paper in section 8, “Development of new therapeutic agents.”

Once again, thank you for giving us the opportunity to strengthen our manuscript with your valuable comments and queries. We have put in the best of our efforts to incorporate your feedback and hope that these revisions persuade you to accept our submission.

Sincerely,

Naoyuki Otani, M.D., Ph.D.

Dokkyo Medical University Nikko Medical Center
632 Takatoku, Nikko city, Tochigi, JAPAN
321-2523
TEL: 0288-76-1515

Reviewer 2 Report

This review article was well written and very informative to the readers of this journal.  I could not find any point to be revised in present form.

Author Response

Thank you for inviting us to submit a revised draft of our manuscript, entitled, “Recent Advances in the Treatment of Pulmonary Arterial Hypertension,” to Pharmaceuticals. We also appreciate the time and effort you and each of the reviewers have dedicated to provide insightful feedback on ways to strengthen our paper. We have attached the revised manuscript incorporating your suggestions for further consideration. We also hope that our edits and the point-by-point responses we provided below satisfactorily address all the issues and concerns you and the reviewers have noted.

Point 1: This review article was well written and very informative to the readers of this journal.  I could not find any point to be revised in present form.

Response 1: Thank you for your peer review.

Sincerely,

Naoyuki Otani, M.D., Ph.D.

Dokkyo Medical University Nikko Medical Center
632 Takatoku, Nikko city, Tochigi, JAPAN
321-2523
TEL: 0288-76-1515

Reviewer 3 Report

Table 1. My suggestion was to make this a central figure and add images of into four quadrants with the same information but in a picture format with arrows at target of action. The non-available therapies could be cited in the text but low yield for the illustration. 

Author Response

Response to Reviewer 3 Comments

Thank you for inviting us to submit a revised draft of our manuscript, entitled, “Recent Advances in the Treatment of Pulmonary Arterial Hypertension,” to Pharmaceuticals. We also appreciate the time and effort you and each of the reviewers have dedicated to provide insightful feedback on ways to strengthen our paper. We have attached the revised manuscript incorporating your suggestions for further consideration. We also hope that our edits and the point-by-point responses we provided below satisfactorily address all the issues and concerns you and the reviewers have noted.

Point 1: Table 1. My suggestion was to make this a central figure and add images of into four quadrants with the same information but in a picture format with arrows at target of action. The non-available therapies could be cited in the text but low yield for the illustration.

Response 1: Thank you for your suggestion. We have created a new figure (Figure 1) demonstrating the various etiologic pathways that cause PAH and have indicated the mechanism of action of the agent classes.

Once again, thank you for giving us the opportunity to strengthen our manuscript with your valuable comments and queries. We have put in the best of our efforts to incorporate your feedback and hope that these revisions persuade you to accept our submission.

Sincerely,

Naoyuki Otani, M.D., Ph.D.

Dokkyo Medical University Nikko Medical Center
632 Takatoku, Nikko city, Tochigi, JAPAN
321-2523
TEL: 0288-76-1515

Round 2

Reviewer 3 Report

Excellent work by the authors improving the manuscript. I believe the changes have made the manuscript better.